# Urinary Biomarkers for Detection of Clinical Endometriosis or Adenomyosis

**DOI:** 10.3390/biomedicines10040833

**Published:** 2022-04-01

**Authors:** Wei-Chun Chen, Chao-Min Cheng, Wan-Ting Liao, Ting-Chang Chang

**Affiliations:** 1Institute of Biomedical Engineering, National Tsing Hua University, Hsinchu 300, Taiwan; lionsmanic@gmail.com (W.-C.C.); vanessaliao@gapp.nthu.edu.tw (W.-T.L.); 2Division of Gynecologic Oncology, Department of Obstetrics and Gynecology, Chang Gung Memorial Hospital at Linkou, Taoyaun 333, Taiwan; tinchang.chang@gmail.com; 3Department of Obstetrics and Gynecology, Chang Gung Memorial Hospital, Keelung 204, Taiwan; 4Department of Obstetrics and Gynecology, New Taipei City Municipal Tucheng Hospital, New Taipei City 236, Taiwan

**Keywords:** endometriosis, urinary biomarkers, alpha-1 antitrypsin, A1AT, vitamin D-binding protein, VDBP, CA125

## Abstract

Endometriosis or adenomyosis can be clinically diagnosed by ultrasound, symptoms, physical examination, and serum CA125. The urinary markers need to be investigated. The aim of our study was to investigate the urinary markers of clinical endometriosis/adenomyosis, and the correlation of serum CA125 was also studied. From the literature, alpha-1 antitrypsin (A1AT), enolase-1, vitamin D binding protein (VDBP), and CA125 in urine and serum were used in our study and measured by enzyme-linked immunosorbent assays (ELISA). Further clinical correlation and detection performance were evaluated. We enrolled 19 normal controls and 33 patients clinically diagnosed with endometriosis/adenomyosis. There were significant differences between studied patients and normal controls, as follows: serum CA125 (130.91 vs. 19.75 U/mL, *p* = 0.004); urinary CA125-creatinine ratio (5.591 vs. 0.254 ng/mg, *p* = 0.028); and urinary VDBP-creatinine ratio (28.028 vs. 7.301 ng/mg, *p* = 0.018). For diagnostic performances, serum CA125 provided the best results, with an area under curve (AUC) of 0.888 (*p* = 0.001) and accuracy of 86.5%. Other excellent results were also found using urinary VDBP (AUC 0.841, *p* = 0.001) and A1AT (AUC 0.722, *p* = 0.011) creatinine ratio. Using three combined biomarkers, serum CA125, urinary VDBP, and A1AT creatinine ratio, provided good detection power (AUC 0.913, *p* = 0.001, sensitivity 90.9%, specificity 76.5%). Double urine markers used in combination with VDBP and A1AT creatinine ratio also provided good diagnostic performance (AUC 0.809, *p* = 0.001, sensitivity 81.8%, specificity 76.5%, accuracy 80%). Further development of non-invasive point-of-care tests using these biomarkers could be a fruitful future endeavor.

## 1. Introduction

Endometriosis, one of most common gynecologic diseases, has a >10% prevalence among women of childbearing age [1]. There are two common types of endometriosis, uterine adenomyosis and ovarian endometrioma, during which ectopic endometriotic cells extend to the uterine myometrium and adnexa, respectively [2]. Endometriosis can lower life quality, with dysmenorrhea, dyspareunia, dyschezia, and hypermenorrhea that may be accompanied with drug overuse, such as painkillers, and severe anemia [3]. Furthermore, endometriosis may also cause infertility and has malignant potential that urges treatment [3]. Diagnosis of endometriosis was previously based on pathology following surgery in past decades [4], but non-surgical detection methods, including ultrasound, serum CA125, and clinical symptoms, were usually applied as part of the clinically diagnostic routine, based on good therapeutic response to pharmacological treatment [5,6]. Additionally, it is also important for medical treatment surveillance to include the performance and analysis of clinical symptoms, ultrasounds, and serum CA125 measurements during follow-up. To enhance surveillance compliance, the development of more convenient, non-invasive diagnostic methods has been important.

Several serum biomarkers, including inflammatory factors, such as IL-1, IL-6, IL-7, IL-12, TGF-beta1, or TNF-alpha, can also be used for the detection or evaluation of endometriosis [7,8,9,10]. Other than serum, biomarkers from peritoneal irrigation fluid, endometrial tissue biopsy, and even urine have been reported [11,12,13]. Urine is a good sample source because it is very convenient, easily available, and collection is non-invasive [12]. Further, urine samples can be kept at room temperature for up to 6 h without degeneration [14,15]. Urinary biomarkers of endometriosis, including Enolase-1 and urinary vitamin D binding protein (VDBP), have previously been reported [16,17]. Alpha-1 antitrypsin (A1AT) in peritoneal fluid [18], as well as higher levels of A1AT isoforms in serum [19], have been also published.

A1AT, an acute-phase protein, encoded by the SERPINA1 gene and related to protease inhibitor activity [20], can protect tissue from inflammation-related proteolytic damage by downregulation of neutrophil activity [21,22], so A1AT may present as a demand of anti-inflammation inside the body. Because endometriosis is a pelvic inflammatory disease, A1AT may be increased in patients with endometriosis [23]. Higher expression of A1AT in serum and peritoneal fluid from patients with endometriosis has been reported [12,18,19]. In 2011, Cho et al. also found increased expression of A1AT in urine samples of patients with endometriosis [16]. Enolase is one of the enzymes involved in the anaerobic glycolysis of tumor cells, e.g., the “Warburg effect” of cancer cells, and it may be increased during tissue inflammation or inflammatory disease states [24]. Enolase-1, can be found in various tissues, including several tumors associated with chronic inflammatory change [24]. In 1995, Walter et al. first detected enolase-1 in the serum of patients with clinically staged endometriosis [25]. In 2013, Yun et al. identified urine Enolase-1 as a targeted marker for detecting endometriosis [17]. VDBP is a plasma protein that can transport vitamin D metabolites to target organs [26] and was a key factor in actin scavenging and the immune system to activate macrophages and recruit neutrophils, monocytes, and fibroblasts during inflammation [27]. Thus, VDBP may also demonstrate increased expression in patients with endometriosis because it is also an inflammatory disease. The VDBP level in peritoneal fluid or serum of endometriosis was inconsistent in different reports [28,29,30]. In 2011, Cho et al. discovered that urinary VDBP could act as a potentially useful diagnostic biomarker for the detection of endometriosis [16].

Notably, the clinical values of both proteins were not obvious and required specific evaluation. Because detection of these biomarkers has demonstrated great utility and because urine samples are more conveniently taken and contribute to diagnostic and follow-up compliance, the aim of our study was to examine urinary CA125, enolase, A1AT, and VDBP in patients with endometriosis/adenomyosis, and evaluate associations to serum CA125.

## 2. Materials and Methods

### 2.1. Patients Collection

In this study, we enrolled healthy controls and patients clinically diagnosed with endometriosis or adenomyosis at Chang Gung Memorial Hospital of Linkou Branch. The clinical diagnosis of endometriosis or adenomyosis was based on clinical criteria including the following: (1) endometriotic symptoms such as dysmenorrhea or hypermenorrhea; (2) uterine adenomyosis or ovarian endometrioma that could be measured by gynecologic ultrasound; (3) endometriosis that could be assessed and evaluated by pelvic examination; (4) no history of previous surgical or medical treatment. Those who did not meet the above criteria were not enrolled. Additionally, those without clinical endometriosis were also judged by the above criteria and were enrolled into control group after confirming no other uterine or adnexal mass. The clinical diagnosis was primarily made by the attending physician and urine collection was additionally performed. This study was conducted following approval by the local ethics committee (IRB No. 201901157B0-1907110010), and all enrolled patients completed informed consent forms. All other clinical data, images, or laboratory results were recorded using an electric chart system.

### 2.2. Sample Management

Based on the above diagnostic principles, we clinically differentiated enrolled patients into endometriosis or non-endometriosis groups. After obtaining informed consent, urine samples were retrieved using a sterile plastic tube. Patient urine was provided only after confirming that the collection day was not during menstruation. The collected urine samples were immediately sent to our laboratory, where they were centrifuged at 1000× *g* for 10 min to remove urine sediments. The supernatant urine was subsequently stored at −80 °C until further use.

Serum CA125 values were also obtained on the same day as urine sample collection because one of the aims of our research was to compare target urinary biomarkers with serum CA125, the current gold standard. Serum CA125 values were determined by the hospital diagnostic medical department.

### 2.3. Enzyme-Linked Immunosorbent Assays (ELISA) Procedure

We examined 4 targeted biomarkers and creatinine level in the urine samples. After thawing frozen urine at room temperature, the concentration of the four biomarkers and urine creatinine was measured via commercially available enzyme-linked immunosorbent assays (ELISAs) according to the manufacturer’s protocols (Human CA125 ELISA Kit, MUC16 (ab274402), abcam, UK; Human alpha 1 antitrypsin ELISA Kit, SERPINA1 (ab108799), abcam, UK; Human Vitamin D Binding Protein ELISA Kit (ab223586), abcam, UK; Human Alpha-Enolase ENO1 ELISA Kit (abx253101), abbexa, UK; Creatinine parameter assay kit (KGE005), R&D systems, Minneapolis, MN, USA). Following the ELISA process, we evaluated absorbance values using a plate reader, and converted the sample concentration using a calculated standard curve. Creatinine normalization was done via division of the obtained 4 urinary biomarkers by urinary creatinine level thereafter.

### 2.4. Statistics

The data were analyzed using SPSS (version 22.0, IBM). To evaluate diagnostic power, receiver operating characteristic (ROC) curves were examined for serum CA125 level, the raw data of above urinary biomarkers, and those with urinary creatinine normalization respectively [31]. Cut-off values were also obtained after ROC curve analysis. To compare the diagnostic accuracy of different sets of these serum and urinary biomarkers, we utilized area under the ROC curve (AUC) to see which combinations were most accurate. The related sensitivity and specificity for selected cut-off points was also investigated. Additionally, subgroup analysis with classifications from different variates was also performed via paired *t*-test or the Mann–Whitney U test. The above analyses were considered significant when the *p*-value was less than 0.05.

## 3. Results

### 3.1. Patient Characteristics

In our study, we collected 52 cases from a single tertiary hospital, beginning in November of 2019. Among the 52 cases, 33 patients were clinically diagnosed with endometriosis or adenomyosis, and another 19 cases were identified as “non-endometriosis/adenomyosis” patients. The clinically diagnostic criteria are described in the previous section. All patient characteristics are provided in Table 1. The median age was 42.9 years old (range 26.1–51.9) of studied patients and 39.2 years old (range 23.7–52.4) for the control group. The median body mass index was 22.7 kg/m^2^ (range 16.3–37.2) and 21.4 kg/m^2^ (range 20.1–31.8) for studied patients and control group, respectively. A total of 42.3% of our cases had no parity history. There were 25 patients with uterine adenomyosis, 4 patients with ovarian endometrioma, and 3 patients with both uterine adenomyosis and ovarian endometrioma. Additionally, the median thickness of adenomyosis was 3.7 cm (range 1.2–10.0), and the median diameter of ovarian endometrioma was 6.9 cm (range 1.5–14.0). Among all patients, 45.4%, 24.2%, and 3.0% of those with endometriosis demonstrated clinical symptoms, including dysmenorrhea, hypermenorrhea, and compression symptoms, respectively.

### 3.2. Median Values and Subgroup Analysis of Targeted Biomarkers

The median values of serum CA125, urine CA125, urine A1AT, urine Enolase-1, and urine VDBP in our cohorts were 31.3 U/mL, 0.08 mg/dL, 47.46 ng/mL, 2.07 ng/mL, and 59.31 ng/mL, respectively. Additionally, urine creatinine was also evaluated, and the median urine creatinine normalized ratio of urine CA125, A1AT, enolase-1, and VDBP were then calculated as 0.0063, 10.17, 0.22, and 8.91 ng/mg, respectively. The above data are provided in list form in Table 2. A basic comparison of the above values, including endometriosis vs. non-endometriosis, ovarian endometrioma vs. uterine adenomyosis, endometrioma vs. non-endometriosis, and adenomyosis vs. non-endometriosis, are provided in Table 3 and Table A1. Further subgroup analysis with different variates, including age, BMI, ovarian endometrioma size, and uterine adenomyosis thickness, were also performed, and the data of all patients with endometriosis, patients with ovarian endometrioma, and adenomyosis cohorts are provided in Table A2, Table A3 and Table A4, respectively.

Only serum CA125 level was significantly different between endometriosis patients and non-endometriotic cohorts (130.91 vs. 19.75 U/mL, *p* = 0.004). Following urine creatinine normalization, both urine CA125-creatinine ratio and urine VDBP-creatinine ratio demonstrated significant differences in median values between patients with endometriosis and non-endometriotic cohorts (urine CA125 ratio: 5.591 vs. 0.254, *p* = 0.028; urine VDBP ratio: 28.03 vs. 7.30, *p* = 0.018). Furthermore, the difference could also be seen in patients with adenomyosis compared to non-endometriotic cohorts (urine CA125 ratio: 7.14 vs. 0.25, *p* = 0.031; urine VDBP ratio: 31.13 vs. 7.30, *p* = 0.033).

### 3.3. Diagnostic Performances of Targeted Biomarkers

The receiver operating characteristic (ROC) curve, to differentiate patients with endometriosis and non-endometriotic cohorts, was calculated and is provided in Figure 1. The detailed data are listed in Table A5 and Table A6. Table A5 discloses the cut-off values and area under curve (AUC) of the ROC curve analysis for serum CA125, urine CA125, urine A1AT-creatinine ratio, urine enolase-creatinine ratio, and urine-VDBP creatinine ratio. Serum CA125, urine VDBP-creatinine ratio, and urine A1AT-creatinine ratio had better AUC values with significance (serum CA125: 0.888, *p* = 0.001; urine VDBP-creatinine ratio: 0.841, *p* = 0.001; urine A1AT-creatinine ratio: 0.722, *p* = 0.011). The cut-off values were 23.75 U/mL, 5.20 ng/mg, and 6.92 ng/mg, respectively.

These three markers were assembled in different combinations to calculate the related ROC curve analysis, and the results are revealed in Table A6. There were good AUC values (0.898 to 0.913, *p* = 0.001) while using three markers in combination (serum CA125 + urine A1AT-creatinine ratio + urine VDBP-creatinine ratio) or (serum CA125 x the value of urine A1AT-creatinine ratio + urine-VDBP-creatinine ratio). Additionally, the combination of serum CA125 with urine VDBP-creatinine ratio also provided excellent AUC value (0.936 to 0.939, *p* = 0.001). Two urinary biomarkers in combination provided slightly lower AUC values, as follows: (1) urine A1AT ratio + urine VDBP ratio (AUC 0.750, *p* = 0.004); or, (2) urine A1AT x urine VDBP ratio (AUC 0.809, *p* = 0.001). However, there were still significant results after calculation.

Table 4 demonstrates the sensitivity, specificity, positive predictive value, negative predictive value, and accuracy for the use of three markers, including serum CA125, urine A1AT-creatinine ratio, and urine VDBP-creatinine ratio. The data for the other five combinations are also listed. Serum CA125 and urinary VDBP ratio both demonstrated good sensitivity, positive predictive values, and accuracy for the clinical detection of endometriosis. The highest accuracy, up to 88%, was found using a combination of serum CA125 and urine VDBP ratio. There were also accuracy rates as high as 80% with 81.8% sensitivity, when combining two urinary biomarkers, such as urine A1AT ratio and urine VDBP ratio.

Figure 1 demonstrates the comparative ROC curves for different biomarkers and combinations with significance. Better AUC values were found when using three-marker combinations. The use of two urinary biomarker combinations provided slightly lower AUC values but still demonstrated good performance for differentiating between patients with endometriosis and non-endometriotic cohorts.

## 4. Discussion

In traditional clinical practice, the gold standard for the diagnosis of endometriosis is pathologic proof from surgical intervention [32], and ectopic endometrial lesions seen during surgery with histologic findings, including the presence of hemosiderin-laden macrophages or endometrial glands [33]. However, because of advances in the development of medical treatment for endometriosis, including Dienogest, Gestrinone, Danazol, Leuprolide acetate, or Levonorgestrel-releasing intrauterine system [34], non-surgical intervention has become more and more popular for patients and physicians. As a result of these advances, the clinical diagnosis of endometriosis without surgical pathology is becoming increasingly frequent [35]. The clinical detection of endometriosis includes gynecologic ultrasound, serum CA125, history taking that includes clinical symptoms, and pelvic examination [36]. Serum sample collection for measurement of serum CA125 is not convenient or without pain and the results typically take several hours. In addition to serum CA125, other potentially useful biomarkers, including peritoneal irrigation, have been mentioned in the literature [18], and several protein isoforms, such as haptoglobin, alpha-1 antitrypsin, S100-A8, and serotransferrin, have been found with higher expression in the peritoneal lavage fluid of patients with endometriosis.

Because the above examination efforts are invasive and inconvenient for patients and physicians, reliable and accurate non-invasive detection methods need to be provided. Urine is a suitable sample source, since it can be easily collected by patients themselves without invasive procedures. Previous research on urinary biomarkers to detect adnexal malignancy have been performed, and several biomarkers, including *N*,*N*-diacetylspermine, HE4, Eosin-derived neurotoxin with CooH-terminal osteopontin fragments, mesothelin, and CA125, have been investigated [37]. Urine CA125 evaluation has a sensitivity that is only 3.3% lower than other methods [38]. Additionally, urinary high-mobility group protein A1 (HMGA1) can be found with higher expression in patients with serous epithelial ovarian cancer [39]. The change in urinary gonadotropin peptide (UGP) level can be used for surveilling patients with gynecologic malignancy after treatment [40]. In 2009, Petri et al. used equalizer bead technology and human urine to detect ovarian cancer and found three urinary biomarkers, including fibrinogen alpha fragment, collagen alpha 1 fragment, and fibrinogen beta NT fragment, to be present at significant levels [41]. Better diagnostic performance was detected when using combined markers, involving multiple biomarkers and serum CA125. For diagnosing endometriosis, previous research discovered two potentially useful urinary biomarkers, Enolase-1 and vitamin D-binding protein (VDBP) [16,17]. In addition to these two biomarkers, there are also a number of proteins related to tissue inflammation, including IL-1, IL-6, IL-8, TNF-alpha, ICAM-1, MMPs, TIMPs, or VEGF, that may prove useful for endometriosis detection [7,8]. Additionally, previous research investigated the role of alpha-1 antitrypsin (A1AT) as a detection tool of endometriosis [18].

A1AT has been identified as a potentially useful biomarker for some time. In published articles, tumor-associated trypsin inhibitor (TATI) has been identified in the urine of patients with ovarian malignancy [42]. Ovarian cystic fluid TATI level can be detected at levels similar to those in urine samples, and at higher levels in mucinous fluid than in serous ovarian cysts [43]. Notably, TATI level measurements in serum or urine have been found useful for follow-up evaluations of patients undergoing cervical adenocarcinoma treatment [44] and patients with solid organ tumors (pancreas, ovary, esophagus, bladder), as well as patients with liver metastasis as a result of colon or breast malignancy [45]. Further, A1AT can also be detected in the tissue of uterine stroma sarcomas, including carcinosarcoma (mixed mullerian tumor), leiomyosarcoma, and endometrial stromal sarcomas. Positive A1AT findings in tissue samples have also been associated with cases of endometrial adenocarcinoma or benign leiomyomas [46]. In our data, there were no significant differences in the median A1AT value between patients with endometriosis and normal controls. Additionally, ROC curve analysis provided an AUC value for urine A1AT ratio of 0.722 with significance (*p* = 0.011). Furthermore, the cut-off value was 5.20 ng/mg, and the accuracy showed 73.1% with sensitivity and specificity as 78.8% and 63.2%, respectively. The above values demonstrated acceptable power for the use of urinary A1AT measurement for the detection of endometriosis.

From the literature, enolase-1 level in urine samples was investigated and found to provide better diagnostic performance when used in conjunction with measurements of serum CA125 (AUC of 0.821, sensitivity of 76.9%, and specificity of 85.0%) [17]. In our study, no obvious significant differences could be detected between urine enolase-1 or the urine enolase-1-creatinine ratio between patients with endometriosis and normal controls. The diagnostic performances indicated an AUC value of 0.487 without significance (*p* = 0.878). This suggests less powerful detection ability for urine enolase-1.

From previous literature, there was significantly increased expression of VDBP in urine samples from patients with endometriosis compared to normal controls (*p* < 0.001) [16], but the diagnostic performance was less than that for serum CA125, with limited sensitivity and specificity of 58% and 76%, respectively. In our research, there were significant differences in the urinary VDBP-creatinine ratio between patients with endometriosis and normal controls (0.280 vs. 0.073, *p* = 0.018). Additionally, patients with adenomyosis also demonstrated a significantly higher urinary VDBP-creatinine ratio compared to normal controls (0.311 vs. 0.073, *p* = 0.033). For diagnostic performances, urinary VDBP demonstrated an AUC value of 0.841 with significance (*p* = 0.001). Although the AUC value for the urinary VDBP-creatinine ratio was not superior to serum CA125 (0.841 vs. 0.888), it still demonstrated potential value as a non-invasive detection biomarker for the detection of clinical endometriosis.

As with the research by Cho et al. [16], our study showed that serum CA125 was still one of the biomarkers with the highest diagnostic performance, including an AUC value of 0.888 with significance (*p* = 0.001), sensitivity of 84.8%, specificity of 89.5%, positive predictive value of 93.3%, and accuracy of 86.5%. Although there were triple biomarker combinations (CA-125 x the results of urinary A1AT-creatinine ratio + urinary VDBP-creatinine reaction, AUC 0.898 to 0.913, *p* = 0.001, accuracy of 86%) and double biomarkers (CA-125 x urinary VDBP-creatinine ratio, AUC 0.939, *p* = 0.001, cut-off value as 193.33, accuracy rate of 88%), that provided better performance, these results primarily depended on the power of serum CA-125, while comparing the data of single markers using urinary A1AT-creatinine ratio (AUC 0.722, *p* = 0.011, cut-off value 5.200, accuracy rate of 73.1%) or urinary VDBP-creatinine ratio alone (AUC 0.841, *p* = 0.001, cut-off value 6.919, accuracy rate of 86%) without obvious differences among each other. The above findings were also compatible with research by Cho et al. [16], who found that the results of combined markers, used in conjunction with serum CA125 measurement, multiplied by urinary VDBP-creatinine ratio was better, but serum CA125 provided the most influence, when comparing urinary VDBP-creatinine ratio (AUC: 0.874 vs. 0.857 vs. 0.678). However, the current study focused on non-invasive methods to detect clinical endometriosis from suspected symptomatic patients, and urinary A1AT-creatinine ratio, urinary VDBP-creatinine ratio and combined data (urinary A1AT + VDBP creatinine ratio), all of which had acceptable performances with AUC values of 0.722, 0.841, and 0.809 with significance, and accuracy rates of 73.1%, 86%, and 80%, respectively. Therefore, both urinary-A1AT and urinary VDBP may be useful tools for non-invasive endometriosis detection in patients with suspected clinical endometriosis.

The one limitation of our current study was that the criteria to differentiate patients with or without endometriosis primarily depended on clinical principles, including clinical symptoms, physical examination, and ultrasound findings, instead of surgical pathologic proof. It was not feasible for us to use surgical classification of endometriosis stage and make a comparison with the above biomarkers in our study. The absence of pathological proof may introduce some bias regarding the accuracy of this biomarker protocol. However, our study may provide additional insight into the potential utility of non-invasive detection biomarkers for clinically suspected endometriosis, and it may provide a conceptually compatible point-of-care test for the non-surgical management of endometriosis. Although the two conditions are not the same [47], we collected patients with endometriosis/adenomyosis for study, since they have similar clinical symptoms and diagnostic methods. The other limitation of our study was the sample size of our normal controls. An increase in enrolled healthy patients may have improved the AUC value and the accuracy for examining urinary A1AT-creatinine ratio. It was difficult to collect balanced and adequate healthy normal controls, since such cases rarely had any need to seek medical consultation.

## 5. Conclusions

In this present study, serum CA125 was identified as the most influential diagnostic performance biomarker for detecting clinical endometriosis or adenomyosis. Additionally, the diagnostic performance for measuring urinary A1AT-creatinine ratio and urinary VDBP-creatinine ratio was further strengthened when used in combination with CA125 measurements. Using these urinary biomarkers in combination to further develop clinical tools for point-of-care testing appears to be a useful and feasible approach for detecting endometriosis.

## Figures and Tables

**Figure 1 biomedicines-10-00833-f001:**
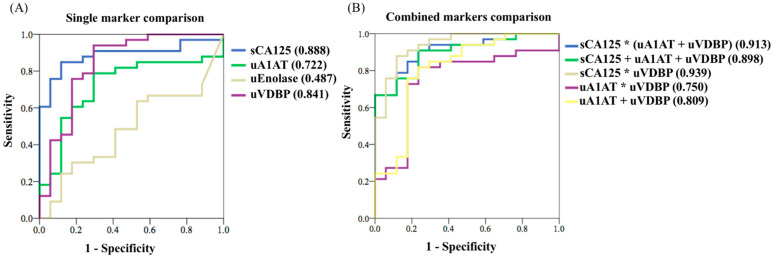
(**A**) The receiver operating characteristic (ROC) curve and area under curve (AUC) values of single examined biomarker. (**B**) The ROC curve and AUV values of combined examined biomarkers. A1AT = alpha 1-antitrypsin; VDBP = vitamin D-binding protein; sCA125 = serum CA125; uA1AT = urine A1AT-creatinine ratio; uVDBP = urine VDBP-creatinine ratio; * = multiply.

**Table 1 biomedicines-10-00833-t001:** Patient characteristics.

	Endometriosis/Adenomyosis	Control (No Endometriosis/Adenomyosis)	*p* Value
Age, median years (range)	42.9 (26.1–51.9)	39.2 (23.7–52.4)	0.076
BMI, median (range)	22.7 (16.3–37.2)	21.4 (20.1–31.8)	0.689
Parity, N (%)			0.310
0	15 (45.5)	7 (36.8)	
1	7 (21.2)	2 (10.5)	
2	8 (24.2)	4 (21.1)	
>2	2 (6.1)	5 (26.3)	
miss	1 (3.0)	1 (5.3)	
Endometriosis status, N (%)			0.001 *
Nil	0	19 (100)	
Ovarian endometrioma	4 (7.7)	0	
Adenomyosis	25 (48.1)	0	
Adenomyosis with endometrioma	3 (5.8)	0	
Pelvic endometriosis	1 (1.9)	0	
Adenomyosis size, median (range) (cm)	3.7 (1.2–10.0)	-	-
Ovarian endometrioma size, median (range) (cm)	6.9 (1.5–14.0)	-	-
Clinical endometriosis symptoms			0.001 *
Dysmenorrhea	15 (45.4)	0	
Hypermenorrhea	8 (24.2)	0	
Compression	1 (3.0)	0	
Hypermenorrhea + Dysmenorrhea	7 (21.2)	0	
Hypermenorrhea + Compression	1 (3.0)	0	
Nil	1 (3.0)	19	

BMI = body mass index; * = data with significance.

**Table 2 biomedicines-10-00833-t002:** The median values of experimented biomarkers.

Total N = 52	Median (Range)
Serum CA125 (U/mL)	31.3 (10.6–1173.5)
Serum Creatinine (mg/dL)	0.56 (0.38–0.88)
Urine Creatinine (mg/dL)	741.17 (5.31–2695.6)
Urine CA125 (ng/mL)	0.0819 (0.0001–1181.8868)
Urine CA125-Creatinine ratio (ng/mg)	0.0063 (0.0001–49.9975)
Urine A1AT (ng/mL)	47.4644 (0.0001–1368.3529)
Urine A1AT-Creatinine ratio (ng/mg)	10.1737 (0.0001–1047.8611)
Urine Enolase-1 (ng/mL)	2.0652 (0.0001–20.8191)
Urine Enolase-1-Creatinine ratio (ng/mg)	0.2236 (0.0001–18.1261)
Urine VDBP (ng/mL)	59.312 (0.344–387.648)
Urine VDBP-Creatinine ratio (ng/mg)	8.9052 (0.0213–194.9118)

A1AT = alpha 1-antitrypsin; VDBP = vitamin D-binding protein.

**Table 3 biomedicines-10-00833-t003:** The median values comparison of examined biomarkers and the urine-creatinine correction in different endometriosis status.

	Urine CA125 (mg/dL)	Urine A1AT (ng/mL)	Urine Enolase (ng/mL)	Urine VDBP (ng/mL)	Serum CA125 (IU/mL)
Emsis/AD	45.53	179.92	3.68	90.657	130.91
No emsis/AD	1.38	98.22	3.95	51.385	19.75
*p* value	0.357	0.280	0.857	0.095	0.004 *
**Urine Creatinine Corrected**	**Urine** **CA125-Creatinine Ratio (ng/mg)**	**Urine** **A1AT-Creatinine Ratio (ng/mg)**	**Urine** **Enolase-1-Creatinine Ratio (ng/mg)**	**Urine** **VDBP-Creatinine Ratio (ng/mg)**	
Emsis	5.591	60.958	1.365	28.028	-
No emsis	0.254	10.404	1.529	7.301	
*p* value	0.028 *	0.233	0.855	0.018	

Emsis = Endometriosis; AD = Adenomyosis; A1AT = alpha 1-antitrypsin; VDBP = vitamin D-binding protein; * = data with significance.

**Table 4 biomedicines-10-00833-t004:** The sensitivity, specificity, positive predictive value, negative predictive value, and accuracy of different biomarkers for endometriosis.

(%)	Cut-Off Value	Sensitivity	Specificity	PPV	NPV	Accuracy
Serum CA125	23.70	84.8	89.5	93.3	77.3	86.5
Urine A1AT-creatinine ratio	5.200	78.8	63.2	78.8	63.2	73.1
Urine-VDBP-creatinine ratio	6.919	93.9	70.6	86.1	85.7	86.0
sCA125 × (uA1AT + uVDBP)	315.847	90.9	76.5	88.2	81.3	86.0
sCA125 + uA1AT + uVDBP	37.229	90.9	76.5	88.2	81.3	86.0
sCA125 × uVDBP	193.330	87.9	88.2	93.5	78.9	88.0
uA1AT × uVDBP	38.602	81.8	76.5	87.1	68.4	80.0
uA1AT + uVDBP	15.651	81.8	76.5	87.1	68.4	80.0

A1AT = alpha 1-antitrypsin; VDBP = vitamin D binding protein; sCA125 = serum CA125; uA1AT = urine A1AT-creatinine ratio; uVDBP = urine VDBP-creatinine ratio; PPV = positive predictive value; NPV = negative predictive value.

## Data Availability

Not applicable.

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
