# Peer review of "Urinary Biomarkers for Detection of Clinical Endometriosis or Adenomyosis"

_biomedicines, 2022, doi:10.3390/biomedicines10040833_

Round 1

Reviewer 1 Report

The problem of non invasive diagnosis of endoemtriosis is important, but the paper is not very inovative. It is good prepared but the clinical importance of conclusions and the use this tests is questinable.

Author Response

Thank you for this comment. We have significantly surveyed the urinary markers in previous literatures, and studied both correlation and diagnostic performance between urine A1AT and serum CA125. We believe that it is meaningful and important to know both correlation and diagnostic performance in clinical endometriosis - potentially developing the noninvasive point-of-care diagnostic tools thereafter.

Reviewer 2 Report

In this paper Chen and colleagues  evaluated alpha-1 antitrypsin (A1AT), enolase-1, vitamin D binding protein (VDBP), and CA125 in urine and serum of endometriosis patients and normal controls. The authors found incresed serum levels of CA125, increased urinary CA125-creatinine ratio and urinary VDBP-creatinine ratio in patients with endometriosis. Morever, they found that serum CA125 provided the best results as diagnostic marker. Using three combined biomarkers, serum CA125, urinary VDBP, and A1AT creatinine ratio provided an impressive detection power suggesting these biomarkers as non-invasive point-of-care tests. 

The manuscript is very interesting but some points need to be improved.

  • Line 51: in order to highlight the inflammatory environment found in endometriosis, IL-12 and TGFB1 should also be added since they play a pivotal role in endometrial cell proliferation and are modulators of immune cells  (PMID: 18295214 and 26708185)
  • 2.3. Enzyme-linked immunosorbent assays (ELISA) procedure: the product code of the ELISA kits used must be reported
  • in order to make the tables easier to read, an asterisk should be added when the difference is statistically significant
  • Figure 1: the AUC values should also be reported in the figure. Moreover, the reader suggest to split the ROC curves in at least 2 graphs to make them easier to interpreter. Line of identity must be shown. 

Reviewer 3 Report

This paper is of clinical interest to validate biomarkers useful in the diagnosis of clinical endometriosis. However, as the authors also point out, the diagnosis of endometriosis in the target patients has not been surgically (histopathologically) diagnosed. If this paper is to be titled "Urinary biomarkers for detection of clinical endometriosis," as the authors have titled it, the study should first be conducted in patients with a surgical diagnosis. The next study should show that the addition of these markers increases the positive diagnosis rate more than the current clinical endometriosis diagnostic method.

Another problem is the mixture of endometriosis and adenomyosis. Currently, endometriosis and adenomyosis are considered different diseases, so cases of adenomyosis or endometriosis complicated by adenomyosis should be excluded (or considered separately, respectively). Also, patient characteristics should be presented separately for the study and control groups, but are presented together here. Another problem with this study is that it is unclear whether the control patients are appropriate controls. There is a conspicuous lack of information that should be included in the paper, such as no description of exclusions for eligible cases and no description of the method for selecting control cases. On the other hand, the paper is unnecessarily long, with lengthy descriptions of well-known endometriosis and validation between subtypes that are not necessary for the aim of this study.

 If these are not corrected, the paper should not be published.

The individual points to be corrected are as follows

Abstract

  1. The aim of the study is not stated.
  2. The methods of the study is not described. 

Introduction

  1. The general well-known description of endometriosis is written too long. The information needed for this study, i.e., problems in clinical endometriosis diagnosis, should be the focus of the description. What is known and not known about the four urinary markers and the literature that indicates their potential usefulness for this diagnosis should be presented. The aims of this study should then be stated.
  2. This is not consistent with the aim and research described here. The aim is not to evaluate associations to serum CA125. It is to examine the urinary markers between the endometriosis group and the control group and whether they can be used for diagnosis. The aim should be stated again with modifications.

Materials and methods

  1. There is no description of how the control cases were selected.
  2. There is no mention of exclusions in the study cases.
  3. Adenomyosis is now considered not endometriosis. Research methods should be modified to treat adenomyosis and endometriosis as separate diseases.
  4. The method of subgroup analysis is not described. All data to be included in the "Results" should be described in the "Methods" section. However, this analysis is not necessary for the purpose of this study.

Results

  1. The characterization of all the subject cases is listed together. Please list them separately in the endometriosis and control groups and indicate that there are no significant differences between the two groups. If there is a significant difference between the two groups, it is inappropriate as a control group.
  2. Subgroup analysis results are useless for this study and should be omitted.

Discussion

  1. Although details about the biomarkers are provided, but these statements need to be consolidated and included in the "Introduction".
  2. The "Discussion" section should describe what was revealed for the first time in the current study, and should include a description of the references needed to discuss it. The "Discussion" is unnecessarily long.

Round 2

Reviewer 3 Report

The paper has been corrected almost appropriately on the points raised by the reviewers and is worthy of publication.  However, the information provided in the "intruduction" is much too long because unnecessary information is cited in this paper. The content of the "INTRUDUCTION" should be limited to information that explains the purpose of this paper. This is an "Original Article", not a "REVIEW", and the contents of the "INTRODUCTION" should be revised again with that understanding.

Author Response

Point 1: However, the information provided in the "intruduction" is much too long because unnecessary information is cited in this paper. The content of the "INTRUDUCTION" should be limited to information that explains the purpose of this paper. This is an "Original Article", not a "REVIEW", and the contents of the "INTRODUCTION" should be revised again with that understanding.

Response 1: Thanks for your opinion. We modified the “INTRODUCTION” section to focus on our study (lines 61-81).